# A Review of Current and Emerging Therapies for Advanced Hepatocellular Carcinoma

Angelica Singh , Sofia Zahid, Ilya Noginskiy, Timothy Pak, Soeb Usta, Marina Barsoum and Uqba Khan *

New York Presbyterian Brooklyn Methodist Hospital—Weill Cornell Medical College, Brooklyn, NY 11215, USA
* Correspondence: uqk9001@med.cornell.edu

**Abstract:** Hepatocellular carcinoma remains a leading cause of cancer-related deaths worldwide. Liver disease including cirrhosis and viral hepatitis remains among the leading causes of hepatocellular carcinoma and despite increased screening, many patients are diagnosed in the advanced stages precluding them from locoregional therapy. Therapeutic agents for advanced hepatocellular carcinoma were limited to Sorafenib for several years; however, with the emergence of molecular targeted therapies including tyrosine kinase inhibitors and vascular endothelial growth factor inhibitors, in addition to immunotherapies, the way hepatocellular carcinoma is treated has changed significantly. In this review, we summarize the key clinical trials that lead to the approval of these agents for systemic treatment of hepatocellular carcinoma and discuss the preferred sequence of treatment options as well as prospective studies for management of hepatocellular carcinoma.

**Keywords:** hepatocellular carcinoma; molecular targeted therapy; immune checkpoint inhibitor; combination therapy



## 1. Introduction

Hepatocellular carcinoma (HCC) is the fourth leading cause of cancer-related deaths in the world according to the GLOBOCAN database. The vast majority of cases occur in the Asia-Pacific region. Although the incidence and mortality rates for many cancers are declining, it is the inverse for HCC. Based on SEER data (from 2006 to 2012), Black and American Indian/Alaska Native patients with HCC have the lowest five-year survival rates [1–5]. HCC typically affects males more often than females, at a ratio of approximately 2.4:1 [6].

Risk factors for HCC include cirrhosis, chronic hepatitis infection (hepatitis B and hepatitis C), heavy alcohol use, aflatoxins, non-alcoholic steatohepatitis, diabetes, smoking, and obesity. Additionally, associated conditions less commonly seen include Wilson's disease, hemochromatosis, alpha-1 antitrypsin, glycogen-storage disease types I and II, porphyrias, and tyrosinemia. Regardless of etiology, cirrhosis remains the leading risk factor in developing HCC. In particular, hepatitis B virus (HBV) carries a risk of 10–25% in developing HCC. Furthermore, when compared to other chronic hepatitis infections, HBV is capable of evolving into HCC without cirrhosis. Patients with persistent HBV infection carry a 223-fold increased risk of developing HCC compared to those not infected. Heavy alcohol consumption is related to liver disease and is more prevalent than hepatitis C virus (HCV) [6].

Multiple professional societies have proposed HCC screening guidelines. The general international consensus is that all patients with liver cirrhosis should be screened for HCC despite the patient's age or etiology of the liver cirrhosis. Those with HBV or HCV along with additional risk factors are also subject to early screening. Generally, it is recommended that patients follow up every 6 months, which was initially proposed due to early studies showing this as the HCC doubling time. Depending on the liver lesion size, follow up may be recommended more often. Across all guidelines, the imaging modality of choice is ultrasonography given that it is inexpensive, more readily available, and non-invasive.

However, ultrasound is limited by operator dependence as well as issues that can arise with visualization. If poor-quality ultrasound is obtained, further imaging could be performed with multiphase contrast-enhanced CT or MRI [7]. Furthermore, a serum AFP > 400 has a high positive predictive value, which is helpful when a diagnosis is unclear despite imaging [6]. Confirmation can be performed with histology [8]. For atypical nodules on CT or MRI, percutaneous biopsy could be performed [6]. Addition of biomarker surveillance, particularly alpha fetoprotein (AFP), is recommended depending on which guidelines are followed [7]. Rather than playing a role in diagnosis of HCC, AFP is helpful in patient follow up and possibly treatment options [8].

Despite efforts to improve surveillance practice, patients typically present once they are symptomatic. Unfortunately, at this time of presentation, patients have progressed to later stages of disease. If the patient is diagnosed in the early stage, there are several effective treatment options available including surgical resection, liver transplantation, and ablation. Patients with locally advanced cancers may be candidates for liver-directed therapies, such as transarterial chemoembolization (TACE), bland embolization, and radioembolization [9]. Only approximately 15–35% of patients that develop HCC are candidates for surgical resection [8]. Historically, patients who were not candidates for loco-regional therapies have had a poor prognosis due to lack of effective systemic therapy options. However, there are several systemic therapies recently approved in advanced and metastatic HCC that has improved patient outcomes. The following article discusses multiple treatment line options for HCC.

## 2. First-Line Treatment

### 2.1. Sorafenib

It has been recognized that RAS and JAK/Stat activation is evident in HCC. This is along with methylation, and subsequent suppression, of tumor-suppressor genes—RASSF1A, CIS, and SOCSI. It was shown that RAS and JAK/Stat inhibitors along with demethylating agents produced an apoptotic response in these HCC tissues [10]. Angiogenesis is heightened in HCC as the receptors for VEGF are overexpressed in HCC endothelial cells [11]. RAS is a tumorigenic signal molecule that activates the RAF/MEK/ERK cascade. A variety of receptor tyrosine kinases stimulate RAS, which induces cell survival and proliferation. Of these, VEGFR is one potent stimulator and is responsible for angiogenesis, which in turn translates into cell growth, migration and branching [12,13].

Sorafenib is a multikinase inhibitor that was developed to block RAF-1. It was shown that not only did this molecule inhibit RAF-1 at relatively low concentrations, but it also had inhibition activity against BRAF, VEGFR, PDGFR, C-KIT and FLT3 at higher concentrations. This made Sorafenib a curiously efficient obstructer of angiogenesis by blocking upstream epi-cellular (VEGFR) and downstream intracellular (RAF-1) targets [13].

Sorafenib's promising role in HCC was initially demonstrated with a multicenter, international, open label, single arm phase II trial that included 137 patients, 73% of whom had Child–Pugh Class A (CP-A). It demonstrated that Time to Progression (TTP) was 4.2 months with overall survival (OS) being 9.2 months. These results were only modestly higher when relating them to a PHASE III trial that assessed doxorubicin vs. PIAF (cisplatin, interferon, doxorubicin, and fluorouracil), which showed 6.8 and 8.7 median OS rates, respectively [14,15].

In the SHARP trial, a phase III trial published in 2008, Sorafenib's role in systemic therapy for advanced, unresectable HCC was effectively demonstrated. 602 patients were assigned to either Sorafenib (400 mg twice daily) or placebo in a 1:1 ratio. Inclusion criteria were ECOG 0-1, CP-A, along with other adequate hematologic and hepatic parameters. Primary outcomes were OS and the time to symptomatic progression; secondary outcomes were the time to radiologic progression and safety [16].

The top three causes of the subjects' HCC were HCV, alcohol use only, and HBV. Median OS was 10.7 months in the Sorafenib group and 7.9 months in the placebo group (HR—0.69; 95% CI, 0.55–0.87; $p < 0.001$). There was a 31% relative reduction in the risk of

death at 1 year as survival rates were 44% and 33% in the Sorafenib group and placebo group, respectively. Radiologic progression was 5.5 vs. 2.8 months (HR—0.58; 95% CI, 0.45–0.74; $p < 0.001$). However, the median time to symptomatic progression did not differ significantly (4.1 vs. 4.9 HR—1.08; 95% CI, 0.88–1.31; $p = 0.77$). Most of the side effects in the Sorafenib group were Grade 1 or 2 in severity—most predominantly diarrhea (39%), fatigue (22%), and hand–foot skin reaction (21%). Grade 3 side effects were limited to diarrhea (8%) and hand–foot skin reaction (8%) [16].

The smaller Asia-Pacific trial, published in 2008, was performed with a 2:1 Sorafenib to placebo ratio. Most of the patients had HBV as the leading source of HCC. It demonstrated a median OS of 6.5 months (95% CI 5.56–7.56) vs. 4.2 months (95% CI, 3.75–5.46) and a TTP of 2.8 (95% CI, 2.63–3.58) vs. 1.4 months (95% CI, 1.35–1.55) for the Sorafenib and placebo groups, respectively. The most common side effects seen were hand–foot skin reaction, diarrhea and fatigue (similarly to the SHARP trial) [17].

Based on these two phase III trials, a pooled exploratory analysis was performed to assess the prognostic factors that favor Sorafenib treatment in HCC. In the Sorafenib group, those that had HCV had a median survival (MS) of 426 days (HR 0.47 [0.32–0.69] compared to placebo) vs. non-HCV patients with 232 days (HR 0.81 [0.66–0.99]); HBV patients had an MS of 198 (HR 0.78 [0.57–1.06] compared to placebo) vs. non-HBV 302 (HR 0.69 [0.55–0.85]). Additionally, in the Sorafenib group, a maximum target lesion size of <6 cm had an MS of 426 days (HR 0.61 [0.46–0.80] compared to placebo) vs. >6 cm lesion with an MS of 222 days (HR 0.75 [0.60–0.94]). Furthermore, this analysis demonstrated that macrovascular invasion, higher neutrophil- to-lymphocyte ratio, and AFP > 200 had poorer OS [18]. This put in perspective the poorer OS in the Asia-Pacific trial compared to the SHARP trial as the population studied in the former trial had mostly HBV as the etiological factor and possessed more extensive tumor invasion. Other sub-variate analyses demonstrated an OS benefit of Sorafenib compared to placebo [19,20].

Hence, in November 2007, Sorafenib became the first FDA-approved systemic treatment for HCC. The current treatment dosage is 400 mg twice a day. Sorafenib remains a reasonable alternate first-line treatment for HCC when other approved first-line therapies are not available, or patients have contraindications to these options.

### 2.2. Lenvatinib

Lenvatinib is a multitargeted tyrosine kinase inhibitor of vascular endothelial growth factor (VEGF) receptors VEGFR1 (FLT1), VEGFR2 (KDR), VEGFR3 (FLT4), fibroblast growth factor (FGF) receptors FGFR1, 2, 3, and 4, platelet-derived growth factor receptor alpha (PDGFRα), KIT, and RET.

A pivotal phase III REFLECT trial assessed approximately 1000 patients from the years of 2013 to 2015 in a 1:1 ratio—randomized to either Lenvatinib (adjusted for body weight) or Sorafenib 400 mg twice a day—to appraise non-inferiority. It was an intention-to-treat analysis with the primary endpoint being OS. Secondary endpoints were progression free survival (PFS), median TTP, and objective response (OR). Study subjects were included if they had CP-A, 1 or more measurable target lesions based on mRECIST, BCLC stage B or C and ECOG of 0-1. Patients with ≥50% liver occupation, obvious invasion of the bile duct, or invasion at the main portal vein were excluded. Non-inferiority was achieved as analysis revealed OS was 13.6 months (95% CI, 12.1–14.9) in the Lenvatinib group vs. 12.3 months (95% CI, 10.4–13.9) in the Sorafenib group (HR 0.92, 95% CI, 0.79–1.06). Secondary endpoints were overall better in the Lenvatinib group (PFS, TTP, and OR).

For Lenvatinib vs. Sorafenib, PFS (7.4 vs. 3.7 months, HR 0.92 [0.79−1.06]), median TTP (8.9 vs. 3.7 months, HR 0.66 [0.57−0.77]), and OR (24.1% vs. 9.2%) were all improved. The most common side effects for Lenvatinib were hypertension, diarrhea, and decreased weight and appetite. In the Sorafenib group, the most common side effects were hand–foot skin reaction, diarrhea, hypertension, and decreased appetite. Fatal adverse events occurred in 11 patients taking Lenvatinib and in four patients taking Sorafenib [21].

A post hoc analysis showed that the Lenvatinib group subjects that developed CP-B later had a shorter OS (6.8 vs. 13.3) compared with CP-A. It was seen that Grade 3 AEs were 71.7% (CP-B) vs. 54.7% (CP-A), and discontinuation rates were 18.3% (CP-B) vs. 7.5% (CP-A) [22]. Another two post hoc analyses reported on patients that received further intervention after this trial. Responders to first-line Lenvatinib who received subsequent anti-cancer medication had a median OS of 25.7 months compared to responders to first line-Sorafenib (receiving post-study anti-cancer medication) having a median OS of 22.3 months. Those that received procedures (either TACE or HIAC) had data showing that responders to first-line Lenvatinib had an OS of 27.2 months (Sorafenib line subjects had too few responders to measure OS) [23,24]. These data show that OS can be expanded for those who respond to Lenvatinib and receive procedural intervention or further medical therapy.

In 2018, Lenvatinib became approved for unresectable HCC, with current NCCN guidelines indicating using this in HCC with CP-A only.

### 2.3. Atezolizumab and Bevacizumab

Immune checkpoint inhibitors have been known to be effective in cancer therapy across several tumor types. PDL-1 and PDL-2 are ligands found on host cells that dampen the process of tumor destruction. These proteins bind to PD-1 receptors on T cells that inhibit their function of surveilling which cells are tumorigenic [25]. It was seen that HCC patients with PDL-1 overexpression had a poorer prognosis with higher rates of recurrence [26]. Atezolizumab is a humanized monoclonal antibody immune checkpoint inhibitor that binds to PD-L1 thus preventing PD-1 activity and reversing T-cell suppression. [27]. Bevacizumab is a monoclonal antibody targeting VEGF inhibiting angiogenesis [28]. A phase Ib GO30140 study compared the dual therapy of Atezolizumab and Bevacizumab against Atezolizumab in patients with untreated unresectable HCC. This study had multiple groups out of which Group F (containing 119 patients with CP-A, half of the patients had HBV infection and 20% HCV infection who were randomized to Atezolizumab plus Bevacizumab vs. Atezolizumab monotherapy in first-line setting) showed similar relative risk (RR 20 vs. 17%), but PFS was increased in the combination therapy (5.6 versus 3.4 months with HR 0.55, $p = 0.11$). One of the limitations to this study was a shorter follow up in the F group (median follow up 6.6 vs. 12.4 months) [29].

A global, multicenter, open-label, phase 3 randomized trial study, IMbrave150, compared Atezolizumab plus Bevacizumab with Sorafenib in patients with unresectable untreated HCC. Patients were assigned in a 2:1 ratio to receive Atezolizumab plus Bevacizumab or Sorafenib. Treatment was continued until unacceptable toxic effects occurred or loss of clinical benefit. Primary endpoints of the study were OS and PFS. Secondary endpoints include ORR, duration of response, time to deterioration of quality of life, physical functioning, and role functioning. Patients had to be evaluated for esophageal varices prior to being enrolled in this clinical trial. Patients excluded were, but not limited to, myocardial infarction within the last 3 months, patients on therapeutic anticoagulation, or co-infection with HCV and HBV.

This study showed improved OS with Atezolizumab plus Bevacizumab as survival at 6 months and 12 months were 84.8% (95% CI 80.9–88.7) and 67.2% (95% CI 61.3–73.1), respectively, vs. 72.2% (95% CI, 65.1–79.4) and 54.6% (95% CI, 45.2–64.0) in the Sorafenib group. PFS was also longer with Atezolizumab plus Bevacizumab, median, 6.8 months (95% CI, 5.7–8.3) vs. Sorafenib, 4.3 months (95% CI, 4–5.6). Serious adverse events occurred more with Atezolizumab-Bevacizumab compared to Sorafenib. Gastrointestinal side effects were the most common reason for discontinuation. Gastrointestinal bleeding was more common in Atezolizumab plus Bevacizumab vs. Sorafenib (7% vs. 4.5%, respectively). The most common side effect was hypertension in Atezolizumab plus Bevacizumab (15%). Other adverse events to recognize were hand–foot skin reaction, transaminitis, increased bilirubin, proteinuria, and thrombocytopenia [30,31].

In addition to OS and improved PFS, this trial also showed improved median time to deterioration in quality of life and functioning in combination Atezolizumab plus Beva-

cizumab combination. The combination also reduced anorexia, diarrhea, fatigue, and pain. This trial demonstrated possible synergistic effects of combination of PD-L1 with VEGF inhibitor in treating advanced unresectable HCC [30,31]. Given efficacy of Atezolizumab plus Bevacizumab, it was approved by the FDA as first-line therapy for unresectable advanced HCC.

### 2.4. Durvalumab and Tremelimumab

Another first-line therapy has recently emerged including Durvalumab plus Tremelimumab. PD-L1 monoclonal antibody, Durvalumab, has been studied with another checkpoint inhibitor targeting CTLA-4, Tremelimumab. Durvalmab was evaluated in 40 patients with CP-A, of which 93% patients had already received Sorafenib. The median OS for all patients was 13.3 months, for HCV positive 19.3 months and for HBV positive 6.3 months [32]. Tremelimumab was also studied in a phase II study which included 21 patients with HCC and chronic HCV infection. This study was performed to test the antitumor and antiviral effect of Tremelimumab in patients with HCC and chronic HCV infection. The results of this pilot trial showed a good safety profile and partial response rate was 17.6% and disease control rate was 76.4% [33].

A randomized, open-label, multicenter, and international phase III HIMALAYA trial showed that combination of Durvalumab plus Tremelimumab reduced the risk of death by 22% in patients with unresectable HCC compared to Sorafenib alone. The study used the STRIDE regimen (Single Tremelimumab REgular Interval Durvalumab; T300+D). The study included 1171 patients who had received no prior systemic therapy and were randomly assigned to three arms: Tremelimumab plus Durvalumab, Durvalumab alone, or Sorafenib alone. The primary endpoint was OS while secondary endpoints were PFS, ORR, duration of response, and safety [34].

There was a 22% lower risk of death with advanced HCC who received Durvalumab plus Tremelimumab compared to Sorafenib (3-year OS = 30.7% with Durvalumab/Tremelimumab vs. 24.7% with Durvalumab and 20.2% with Sorafenib). The ORR was 20.1% compared to 17% for Durvalumab and 5.1% for Sorafenib (HIMALAYA). Treatment-related adverse effects occurred in 25.8% of patients on Durvalumab/Tremelimumab combination, 12.9% of patients on Durvalumab, and 36.9% of patients on Sorafenib. One important distinction is that no esophageal varices hemorrhage occurred in this study and the study population did not require an EGD prior to treatment [34]. This combination treatment has yet to obtain FDA approval.

## 3. Second-Line Treatments

### 3.1. Nivolumab

Nivolumab is a IgG4 monoclonal antibody that binds the PD-1 receptor on T cells, thereby enhancing checkpoint signaling [35]. Checkmate 040 was a phase I/II trial that was conducted to evaluate safety response to Nivolumab therapy. Trial subjects had either HCV or HBV, CP-A/CP-B, histologically proven HCC, and allowed previous Sorafenib therapy (although some refused or could not tolerate other therapy). There were two groups in the study: the dose escalation group had 48 people and the dose expansion group had 214 people. Those receiving the dose escalation group had 0.1–10 mg/kg every two weeks and the dose expansion group had 3 mg/kg every two weeks. The dose escalation group hoped to determine maximally tolerated dosing. The objective response rate was 20% (95% CI 15–26), and median PFS being 4 months, in patients treated in the dose-expansion phase and 15% (95% CI 6–28) in the dose-escalation phase. 12 out of 48 (25%) subjects in the dose escalation group, had Grade 3 and above adverse side effects, determined to be independent of dosing. In available subjects for retrospective analysis of biopsied tissue, objective responses were observed in nine (26%) of 34 patients with PD-L1 expression on at least 1% of tumor cells and in 26 (19%) of 140 patients with PD-L1 on less than 1% of tumor cells [35].

Comparing Nivolumab with Sorafenib, the Checkmate 459 Phase III trial had 743 patients receiving the above therapies in a 1:1 ratio. This was an intention to treat analysis, with the primary endpoint being OS. Secondary endpoints were ORR and PFS. Nivolumab dosing was 240 mg IV every two weeks and Sorafenib 400 mg PO BID was used. Nivolumab had a median OS of 16.4 months and Sorafenib had a median OS of 14.7. Even though statistically significant survival benefit was not reached at $p < 0.0419$, Nivolumab showed a rather more tolerable side effect profile. Additionally, patients had a superior complete response rate with 15% in the Nivolumab group and 7% in the Sorafenib group. Hand–foot skin reaction was reported in <1% of the Nivolumab subjects and in 14% of the Sorafenib subjects. Other comparisons were AST increase (6% vs. 4%) and hypertension (0 vs. 7%) in Nivolumab vs. Sorafenib, respectively. Serious treatment-related adverse events were reported in 43 (12%) patients receiving Nivolumab and 39 (11%) patients receiving Sorafenib [36]. In September 2017, Nivolumab was approved by the FDA for treatment of HCC for patients that did not respond to Sorafenib. However, Nivolumab was later withdrawn from USA market as a monotherapy for patients with HCC who were previously treated with Sorafenib.

### 3.2. Nivolumab + Ipilimumab

Nivolumab is a human monoclonal antibody that targets the programmed cell death 1 receptor (PD-1). Ipilimumab is an inhibitor of cytotoxic T-lymphocyte-associated protein 4 (CTLA-4). When combined, this therapy targets two different immune checkpoints. Nivolumab's efficacy as monotherapy was first studied in a phase I/II trial, CheckMate 040. Patients included those with advanced HCC and CP-A or CP-B. These patients had been treated with Sorafenib previously or had been intolerant to therapy. Analysis of the entire cohort showed objective response rates of 23% and 16–19% in Sorafenib-naive and Sorafenib-experienced patients, respectively [35]. Nivolumab monotherapy received FDA approval in 2017 as a second-line therapy for HCC; however, it was rescinded in 2021.

In phase I/II CheckMate 040 trial, Nivolumab and Ipilimumab combined therapy was also studied for efficacy. It was an open-label, multicohort, and multicenter study. Data were collected starting from January 2016 until January 2019 and median follow up was 30.7 months. Participants were recruited from 31 centers across Europe, Asia, and North America. Cohort 4 included 148 Sorafenib-treated patients who were randomly assigned to one of three groups. Of the 148 patients, 120 were male (81%). Group one (50 participants) consisted of Nivolumab 1 mg/kg plus Ipilimumab 3 mg/kg every three weeks for four doses, which was followed by Nivolumab 240 mg every two weeks. Group two (49 participants) used Nivolumab 3 mg/kg with Ipilimumab 1 mg/kg every three weeks for four doses and followed by Nivolumab 240 mg every two weeks. Group three (49 participants) was treated with Nivolumab 3 mg/kg every two weeks plus Ipilimumab 1 mg/kg every six weeks. Multiple primary endpoints were assessed, including safety, tolerability, and objective response rate. Across the three groups, group one, which included Nivolumab 1 mg/kg plus Ipilimumab 3 mg/kg every three weeks for four doses, followed by Nivolumab 240 mg every two weeks, showed the best outcomes. The objective response rate in group one was 32% (95% CI, 20–47%). Group two's objective response rate was 27% (95% CI, 15–41%), and group three was 29% (95% CI, 17–43%). Four patients showed complete response and median OS was 22.2 months (with the other two groups 12.5 and 12.7 months). Median duration of response was 17.5, 22.2, and 16.6 months for each of the three groups, respectively [37].

Additionally, approximately 20% of this cohort demonstrated immune-mediated adverse events. The median time of onset was 1.3 months. These side effects included rash, adrenal insufficiency, hypothyroidism or thyroiditis, colitis, pneumonitis, and infusion-related reactions. 70% of these patients received high-dose glucocorticoids and complete resolution occurred in 70% [37].

Previously, Nivolumab monotherapy, as studied in CheckMate 040, showed an objective response rate of 14%. These advanced HCC patients previously treated with Sorafenib showed a median survival of 15.1 months with monotherapy with manageable safety [35].

In March 2020, combination Nivolumab and Ipilimumab was approved for HCC patients previously treated with Sorafenib. This combination is to be given every three weeks for four doses followed by Nivolumab 240 mg every two weeks or 480 mg every four weeks.

### 3.3. Pembrolizumab

Pembrolizumab is another monoclonal antibody targeted against PD-1. This second-line therapy is also an option for patients who have failed first-line therapy with Sorafenib. Its effects were first recognized in the phase II Keynote-224 trial, in which patients had advanced HCC that failed Sorafenib and were CP-A. 47 medical centers and hospitals from 10 countries participated. 104 patients were enrolled in the study starting January 2016 and data were collected up until February 2018. Intravenous Pembrolizumab 200 mg was given every three weeks for approximately two years. Treatment was stopped prior to two years if disease progressed, side effects were unacceptable or if the patient and/or investigator decided to withdraw. The median duration of treatment was 4.2 months. An objective response rate of 17% (95% CI 11–26%) was seen in 18 of 104 patients. Stable disease was seen in 44% in patients (46 patients), 16% partial responses (17 patients) and 1% complete (1 patient). Of the 104 patients, 76 patients (73%) had treatment-related adverse events. Serious side effects were seen in 16 (15%) patients and one death from ulcerative esophagitis secondary to treatment [38].

Phase III KEYNOTE-240 trial further supported Pembrolizumab as an alternative PD-1 inhibitor for second-line therapy. This study evaluated the efficacy and safety of the medication. This double-blinded study included 413 patients that were assigned randomly to Pembrolizumab or placebo on a 2:1 basis. Patients on Pembrolizumab showed higher objective response rate (18.3% vs. 4.4%) as well as complete responders (6 vs. none). [39].

In the Pembrolizumab arm, 269 patients (96.4%) had adverse events. Of this group, 48 patients discontinued treatment due to adverse events, which included ascites in 12 patients (4.3%) and increased AST and serum bilirubin in four patients (1.4%). In comparison, in patients receiving placebo, 121 patients (90.3%) had an adverse event. Three patients (2.2%) had ascites and one (0.7%) patient had increased AST and serum bilirubin. Treatment was interrupted for 84 patients (30.1%) on Pembrolizumab due to elevated bilirubin in 15 patients (5.4%) and elevated AST levels in 13 patients (4.7%). 21 patients (15.7%) in the placebo group had an interruption in treatment for elevated bilirubin in 5 patients (3.7%) and elevated AST in 4 patients (3%). Death due to adverse events was seen in 7 patients (2.5%) on Pembrolizumab and 4 patients (3%) on placebo. Common adverse events in the Pembrolizumab group included increased AST and serum bilirubin levels, fatigue, and pruritus. In the placebo group, adverse events were elevated AST levels, fatigue, and cough. Immune-mediated adverse events most seen were hypothyroidism, and pneumonitis in 51 patients (18.3%) of the Pembrolizumab group (versus 11 patients or 8.2% in placebo). Additionally, 10 patients (3.6%) in this group showed immune-mediated hepatitis events, which 90% resolved, as well as no hepatitis B or C viral flares. Steroids were given to 23 patients (8.2%) in the Pembrolizumab group and one (0.7%) in the placebo group [39]. Based on this trial, Pembrolizumab 200 mg IV or 400 mg IV every three or six weeks, respectively, was approved in patients previously treated with Sorafenib.

### 3.4. Ramucirumab

Ramucirumab is a recombinant monoclonal antibody which inhibits vascular endothelial growth factor receptor 2 (VEGFR2). It is part of the immunoglobulin G subclass 1 (IgG1) and is a recombinant monoclonal antibody that binds to VEGFR-2 and blocks activation. In HCC, high AFP is seen with increased vascular endothelial growth factor (VEGF) and VEGFR-2 expression.

The REACH trial, a phase I/II, randomized and double-blinded study, investigated Ramucirumab in 42 patients with advanced HCC after failing Sorafenib. Ramucirumab was studied against a placebo. The median OS was 9.2 months in Ramucirumab plus

placebo versus 7.6 months in only placebo. It also showed an objective response of 10% and 12 months of median OS. However, it was unable to show significant survival advantage. When the data were further analyzed, patients that had a higher level of AFP (>400 ng/mL) at the time of diagnosis showed a median survival of 7.8 months versus 4.2 months, suggesting a potential for greater survival and further studies [40,41].

REACH-2, a randomized phase III trial, looked at 292 patients with HCC, CP-A or better and at least a serum AFP of 400 or more. These patients additionally had progression of disease or had failed Sorafenib therapy. Patients were randomly assigned 2:1 to either Ramucirumab 8 mg/kg intravenously or placebo with best supportive care every two weeks. The study continued until disease progression or toxicity that was not acceptable. The primary endpoint was OS and secondary objectives were PFS, ORR, and safety. Significant results were found in OS and PFS. OS was 8.5 months compared to the placebo, which was 7.3 months (HR 0.710; 95% CI 0.531; *p* = 0.0199). PFS was a median of 2.8 months vs. 1.6 months in the placebo group (HR 0.452; 95% CI 0.339, 0.603; *p* < 0.0001) [40].

In a pooled analysis between these trials, treatment-related adverse events commonly seen with Ramucirumab were mostly mild to moderate, which included peripheral edema, anorexia, nausea, fatigue, diarrhea, abdominal pain, and headache. Treatment emergent adverse reactions included hypertension and hyponatremia [42].

Additionally, three patients expired from drug-related complications including acute kidney injury, hepato-renal syndrome, and renal failure. Overall, in comparison to first-line agents such as Sorafenib, the toxicity profile for Ramucirumab was shown to be more favorable [42]. REACH-2 concluded that in HCC patients with AFP 400 ng/mL or more, there is an increased survival benefit and reduced risk of death (29%) [40]. In May 2019, the FDA approved Ramucirumab as second-line monotherapy option for HCC patients with AFP 400 ng/mL or greater that failed Sorafenib as first-line therapy.

### 3.5. Regorafenib

Regorafenib is an orally active multikinase inhibitor that targets angiogenic, stromal, and oncogenic receptor tyrosine kinases that ultimately lead to halting overall tumor growth. Being almost identical in molecular structure to Sorafenib, its specific mechanism of action is inhibition of VEGF receptors 1-3, KIT, PDGFR-alpha, PDFR-beta, RET, FGFR1 and 2, TIE2, DDR2, TrkA, Eph2A, RAF-1, BRAF, BRAF V600E, SAPK2, PTK5, and Abl [43].

The phase III RESORCE trial enrolled patients with CP-A and Barcelona Clinic Liver Cancer Stage Category B or C HCC, with HCC progression on Sorafenib, where participants were randomized to receive 160 mg of Regorafenib once daily for the first 21 days of each 28 cycle until disease progression or overwhelming toxicity. This study design prevented dropouts from adverse reactions to medication as well as minimizing the effect of post-trial treatment after progressive disease on Regorafenib. Regorafenib was found to have beneficial outcomes during the overall stages of the RESORCE trial, as it was associated with greater OS of 10.6 months vs. 7.8 months in placebo (HR, 0.63; 95% CI, 0.5–0.79; *p* <0.001) as well as higher rates of antitumor response and disease control (7). In addition, Regorafenib was associated with minor overall side effects including hypertension, hand foot skin reaction, fatigue, and diarrhea. Compared to placebo, it has prolonged survival and was also associated with a decrease in death by a substantial 37%. In addition, greater PFS (3.1 months vs. 1.5 months; HR, 0.46; 95% CI, 0.37–0.56; *p* < 0.001), TTP by mRECIST (3.2 months vs. 1.5 months; HR, 0.44; 95% CI, 0.36–0.55; *p* <0.001), disease control (65% vs. 36%; *p* < 0.001), and objective rate response (11% vs. 4%; *p* = 0.005) were also seen [44]. Additional evidence from the RESORCE trial shows that Regorafenib offers clinical benefit independent of the last Sorafenib treatment [45]. In 2017, the FDA approved Regorafenib for HCC in patients that have progressed on Sorafenib. Recommended dosage is 160 mg orally, once daily, for the first 21 days of each 28-day cycle.

### 3.6. Cabozantinib

Cabozantinib is a potent inhibitor of pro-invasive receptor tyrosine kinases (RTKs), including AXL, FLT-3, KIT, MER, MET, RET, ROS1, TIE-2, TRKB, TYRO3, and VEGFR-1, -2, and -3 [46]. Cabozantinib first showed its benefits to those suffering from progression of HCC (along with those on second- and third-line treatment previously on Sorafenib) in the phase III CELESTIAL trial. The CELESTIAL trial was a randomized, double-blind, placebo-controlled trial, where patients with advanced HCC were randomized to receive 60 mg of Cabozantinib daily versus placebo until disease progression or overwhelming toxicity. The patients that were in the randomized clinical trial had CP-A. In addition, randomization was stratified by those with hepatitis B and/or C, geographic location, and presence of extrahepatic spread. Among the study's candidates, 7.6% of the patients in the trial already received more than one previous line of treatment. Median OS and PFS were greater in those that received Cabozantinib versus those on placebo (10.2 months and 5.2 months vs. 8.0 and 1.9 months, respectively) with hazard ratio (HR), 0.76; 95% CI, 0.63–0.92; $p = 0.005$ for OS and HR, 0.44; 95% CI, 0.36–0.52; $p < 0.001$ for PFS. Side effects were noted to be primarily that of hand–foot skin reaction, hypertension, increased aspartate transferase, fatigue, and diarrhea. These side effects were seen in 68% of those that received Cabozantinib and 36% of those that received placebo [47]. Additional analyses revealed that Cabozantinib had improvement in outcomes versus placebo across a wide range of baseline alpha-fetoprotein levels [48]. In 2019, the FDA approved Cabozantinib for patients with Child–Turcotte–Pugh class score A who had HCC disease progression on Sorafenib as second-line and third-line therapy options. Cabozantinib comes in tablets of 20 mg, 40 mg, and 60 mg, and its recommended dose is 60 mg orally once a day.

### 4. How to Sequence Therapy

Treatment options for advanced hepatocellular carcinoma have evolved remarkably in the last few years as listed in Table 1; however, there is no consensus or algorithm on how to sequence therapy. Patients with preserved liver function as well as performance status are candidates for systemic treatment of advanced hepatocellular carcinoma. In the first-line setting, our recommendation is Atezolizumab plus Bevacizumab in patients without contraindication to immune checkpoint inhibitors or anti-angiogenic therapy. Patients with history of autoimmune disease, untreated or incompletely treated varices (esophageal or gastric) or with high bleeding risk are not candidates for first-line Atezolizumab plus Bevacizumab [30]. In patients who are unable to receive Atezolizumab/Bevacizumab, the combination of Tremelimumab plus Durvalumab has emerged as a promising first-line alternative based on the results of the HIMALAYA trial [34]. In patients who are unable to receive checkpoint inhibitors, remaining options are Sorafenib or Lenvatinib in the first line. Our recommendation is Lenvatinib over Sorafenib based on the REFLECT trial which showed a favorable tolerability profile as well as increased progression free survival when compared with Sorafenib [21]. Sorafenib was the first approved agent for systemic therapy in advanced hepatocellular carcinoma and is still widely utilized in this setting.

Given the increasing number of options available in the first-line setting, subsequent lines of therapy depend on initial treatment received by the patient. Patients who received first-line therapy with Atezolizumab/Bevacizumab or Tremelimumab/Durvalumab, our recommendation is TKI therapy. These patients may receive treatment with Lenvatinib, Regorafenib or Cabozantinib. For patients who received TKI therapy with Sorafenib or Lenvatinib in the first line, our recommendation is treatment with immune checkpoint inhibitor, either alone or in combination. For patients with prior TKI therapy and who are not candidates for immune checkpoint inhibitors, options include Regorafenib, Ramucirumab, and Cabozantinib. Ramucirumab is recommended in the second-line setting for patients with AFP greater than or equal to 400 (Figure 1).

**Table 1.** FDA approved therapies for advanced hepatocellular carcinoma.

| | | | Approved Advanced HCC Systemic Therapies | | | |
|---|---|---|---|---|---|---|
| Line of Therapy | Drug | Trial | Total Number (N) | Patient Characteristics | Overall Survival (Months) | Progression Free Survival (Months) |
| First Line | Atezolizumab + bevacizumab vs. sorafenib | IMBrave150: Phase III | 336 vs. 165 | Unresectable, CTP-A, ECOG 0-1, no prior systemic therapy | 67.2% (95% CI, 61.3–73.1) vs. 54.6% (45.2–64) at 12 months | 6.8 (95% CI, 5.7–8.3) vs. 4.3 (95% CI, 4.0–5.6) |
| First Line | Lenvatinib vs. Sorafenib | REFLECT: Phase III | 478 vs. 476 | Unresectable without invasion of main portal vein or biliary tree, CTP-A, ECOG 0-1, no prior systemic therapy | 13·6 months (95% CI, 12.1–14.9) vs. 12.3 months (10.4–13.9) NON INFERIOR | 7.4 (95% CI, 6.9–8.8) vs. 3.7 months (95% CI, 3.6 to 4·6 months) ORR: 24.1% vs. 9.2% |
| First Line | Sorafenib vs. placebo | SHARP: Phase III | 299 vs. 303 | Advanced, CTP-A, ECOG 0-1-2 | 10.7 months (95% CI, 9.4–13.3) vs. 7.9 (95% CI, 6.8–9.1) | PFS not assessed Time to radiologic progression: 5.5 (4.1–6.9) vs. 2.8 (2.7–3.9) |
| Second Line | Pembrolizumab vs. Placebo [OS and PFS not reached] | KEYNOTE-240: Phase III | 278 vs. 135 | Progression or intolerance to sorafenib, advanced, CTP-A, ECOG 0-1 | 13.9 months (95% CI, 8.3—4.1) vs. 10.6 months (8.3–13.5) | 3.0 (95% CI, 2.8–4.1) vs. 2.8 (95% CI, 2.5–4.1) |
| Second Line | Ramacirumab vs. Placebo | REACH-2: Phase III | 197 vs. 95 | BCLC-B or C, CTP-A, ECOG 0-1, AFP >399, previous sorafenib use (first use of biomarker) | 8.5 months (95% CI, 7.0–10.6) vs. 7.3 months (5.4–9.1) | 2.8 (95% CI, 2.8–4.1) vs. 1.6 (95% CI, 1.5–2.7) |
| Second Line | Pembrolizumab | KEYNOTE224: Phase II | 104 | Intolerant of or previously treated with Sorafenib, Advanced, CTP-A, ECOG 0-1 | OS not studied Objective Response: 17%; 95% CI, 11–26 | PFF Not studied |
| Second Line | Cabozantinib vs. placebo | CELESTIAL: Phase III | 470 vs. 237 | Advanced HCC that had received at least one, but up to two lines of previous systemic therapy, including sorafenib, CTP-A, ECOC 0-1 | 10.2 months (95% CI, 9.1–12.0) vs. 8.0 (95% CI, 6.4 to 9.4) | 5.2 (95% CI, 4.0–5.5) vs. 1.9 (95% CI, 1.9–1.9) |

**Table 1.** *Cont.*

| Approved Advanced HCC Systemic Therapies | | | | | | |
|---|---|---|---|---|---|---|
| **Line of Therapy** | **Drug** | **Trial** | **Total Number (N)** | **Patient Characteristics** | **Overall Survival (Months)** | **Progression Free Survival (Months)** |
| Second Line | 1.Nivolumab 2. Nivolumab plus ipilimumab | CheckMate040: Phase II | (1)262 (2)148 | Advanced, CTP-A or B7, ECOG 0-1, with or without HBV and HCV, previous sorafenib allowed | (1) 9 month OS: 74% (95% CI, 67–79) ORR 20% (95% CI, 15–26) (2) OS 22.8 months (95% CI 9.4–NA) ORR 32% (95% CI 20–47) | |
| Second Line | Regorafenib vs. placebo | RESORCE:Phase III | 379 vs. 194 | Advanced, Progression on sorafenib, CTP-A, ECOG 0-1 | 10.6 months (95% CI, 9.1–12.1) vs. 7.8 (95% CI, 6.3–8.8) | 3.1 (95% CI, 2.8–4.2) vs. 1.5 (1.4–1.6) |

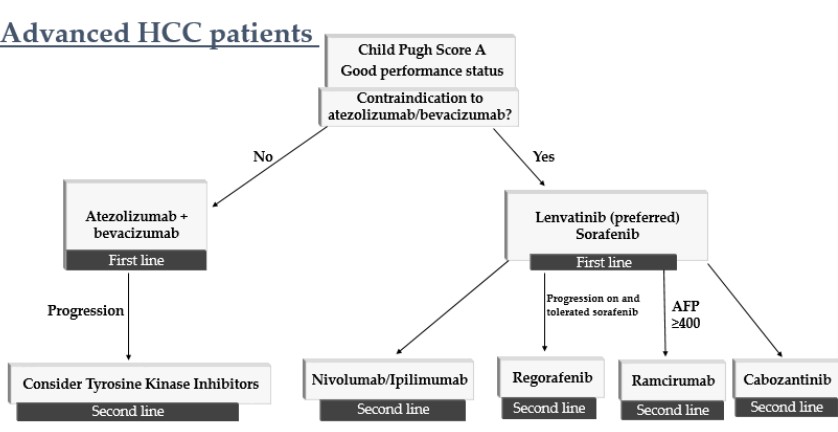

**Figure 1.** Algorithm on how to sequence therapy in patients with advanced hepatocellular carcinoma.

In patients with persevered liver function and excellent performance status, Atezolizumab/Bevacizumab is preferred in first line. If patients are not candidates for immunotherapy or therapy with VEGF inhibitor, options in the first-line include Lenvatinib and Sorafenib. In the second line, systemic therapy options include Cabozantinib, Regorafenib, Ramucirumab (if AFP $\geq$ 400) and if patients have not received immunotherapy in the first line, Nivolumab, Pembrolizumab or combination Nivolumab Ipilimumab may be considered.

## 5. Advanced HCC in Patients with Child–Pugh Class B and C

Many patients diagnosed with advanced hepatocellular carcinoma unfortunately have compromised liver function limiting treatment options. Most approved medications were only studied in patients with preserved liver function (CP-A). In patients with CP-A or CP-B with a score of 7, patients may receive Sorafenib though adverse events often leading to discontinuation of therapy were higher in patients with CP-B than CP-A [49]. In patients with CP-B with a score of 7 or less, Nivolumab may be considered if the patient does not have any contraindication to immune checkpoint inhibitor [35].

In patients with underlying cirrhosis and advanced hepatocellular carcinoma, particularly Child–Pugh Class C, supportive measures are recommended as these patients are unlikely to be able to tolerate or benefit from systemic therapy [50].

## 6. Ongoing and Future Clinical Trials for HCC

There are numerous ongoing clinical trials for management of HCC at every stage that will likely change the landscape of treatment. After resection or ablation, a population of patients remain at high risk of disease recurrence and there are no approved regimens in the adjuvant setting. Clinical trials are also underway to evaluate the combination of locoregional therapy in conjunction with systemic therapy. Given the effectiveness immunotherapy has shown combined with anti-angiogenic therapy in HCC, numerous clinical trials (outlined in Table 2) aim to explore combination regimens of immunotherapy, CTLA4 inhibitors, and tyrosine kinase inhibitors.

Currently there are no approved therapies in the adjuvant setting for patients who are at high risk of recurrence of HCC after curative resection or ablation. There are four ongoing phase 3 trials to evaluate the role of immunotherapy in the adjuvant setting. Checkmate-9DX (NCT03383458) aims to evaluate if adjuvant Nivolumab will improve recurrence-free survival in patients who are high risk of recurrence after curative hepatic resection or ablation [51]. EMERALD-2 (NCT03847428) evaluates the safety and efficacy of Durvulamab monotherapy and Durvulamab plus Bevacizumab as adjuvant treatment in patients who are at high risk of recurrence after undergoing curative treatment for HCC with either hepatic resection or ablation with patients. KEYNOTE-937 (NCT03867084) evaluated the safety and efficacy of adjuvant Pembrolizumab in patients who have underwent surgical

resection or ablation with complete radiological response. Checkmate9DX, EMERALD-2, KEYNOTE-937 are all phase 3, randomized, double-blind, placebo-controlled studies with primary endpoints of recurrence-free survival [51–53]. IMbrave050 (BCT04102098) is a phase 3 trial which aimed to evaluate the efficacy and safety of Atezolizumab plus Bevacizumab in the adjuvant setting compared with active surveillance in patients who are at high risk of disease recurrence with a primary endpoint of recurrence-free survival [54].

**Table 2.** Ongoing clinical trials for treatment of hepatocellular carcinoma.

| Trial Identifier | Phase | Setting | Child–Pugh | Treatment Arms | Primary Endpoint |
|---|---|---|---|---|---|
| Checkmate-9DX NCT03383458 | 3 | Adjuvant | Child–Pugh A | Nivolumab vs. placebo | Recurrence-free survival |
| EMERALD-2 NCT03847428 | 3 | Adjuvant | Child–Pugh A | Durvalumab + Bevacizumab Durvalumab + Placebo Placebo + Placebo | Recurrence-free survival |
| KEYNOTE-937 NCT03867084 | 3 | Adjuvant | Child–Pugh A | Pembrolizumab vs. Placebo | Recurrence-free survival |
| Imbrave 050 BCT04102098 | 3 | Adjuvant | Child–Pugh A | Atezolizumab + Bevacizumab vs. Active surveillance | Recurrence-free survival |
| EMERALD-1 NCT03778957 | 3 | Intermediate | Child–Pugh A—B7 | TACE + Durvalumab + placebo vs. Durvalumab + Bevacizumab + TACE vs. Placebo + TACE | Progression free survival |
| LEAP-012 NCT04246177 | 3 | Intermediate | Child–Pugh A | Lenvatinib + Pembrolizumab + TACE vs. Oral placebo + IV placebo + TACE | Progression free survival |
| Checkmate-74W NCT04340193 | 3 | Intermediate | Child–Pugh A | Nivolumab + Ipilimumab + TACE vs. Nivolumab + Placebo + TACE vs. Placebo + Placebo + TACE | Time to TACE Progression |
| RATIONALE-301 NCT03412773 | 3 | Advanced | Child–Pugh A | Tislelizumab vs. Sorafenib | Overall survival |
| Checkmate 9DW NCT04039607 | 3 | Advanced | Child–Pugh A | Nivolumab + Ipilimumab vs. Sorafenib/Lenvatinib | Overall survival |
| LEAP-002 NCT03713593 | 3 | Advanced | Child–Pugh A | Lenvatinib + Pembrolizumab vs. Lenvatinib + Placebo | Progression free survival Overall survival |
| COSMIC312 NCT03755791 | 3 | Advanced | Child–Pugh A | Cabozantinib + Atezolizumab vs. Sorafenib vs. Cabozantinib | Progression free survival Overall survival |

In patients with intermediate stage HCC, curative treatment may not be an option; however, locoregional therapy with transarterial chemoembolization (TACE) can achieve tumor response. Patients who undergo TACE therapy commonly progress and recur, often within one year of treatment [55]. There are currently multiple phase 3 studies dedicated to evaluating the role of systemic therapy in combination locoregional therapy with TACE in patients who are not candidates for curative therapy. EMERALD-1 (NCT03778957) evaluates TACE treatment in combination with Durvulamab monotherapy or TACE treatment in combination with Durvulamab with Bevacizumab compared with TACE therapy alone [55].LEAP-012 aims to evaluate Lenvatinib plus Pembrolizumab in combination with locoregional TACE therapy compared with TACE alone [56]. Checkmate 74W has three arms and evaluates TACE in combination with checkpoint inhibitors, Nivolumab and Ipilimumab. Patients are randomized to receive Nivolumab plus Ipilimumab plus TACE, Nivolumab plus TACE, and TACE alone [57]. EMERALD-1, LEAP-012, Checkmate 74W are all phase 3, randomized, placebo controlled, trials with primary endpoints of EMERALD-1 and LEAP-012 being PFS and the primary endpoint of Checkmate 74w being time to TACE progression [55–57].

There have been recent advances in the first-line setting in recent years with TKI and immunotherapy options. Tiseleizumab is a PDL1 inhibitor that has shown favorable results in Phase 1a/Ib trials in patients with advanced solid malignancy. The Rationale-301 study (NCT03412773) compares tislelizumab to Sorafenib as a first-line treatment in patients with unresectable HCC [58]. Nivolumab plus Ipilimumab is currently approved in the second-line setting for patients with advanced HCC [37]. CheckMate-9DW (NCT04039607) evaluates the combination of Nivolumab and Ipilimumab in the first-line setting compared to Sorafenib or Lenvatinib in patients with advanced HCC [59]. There are ongoing phase 3 trials to evaluate combination of immunotherapy with TKI in the front line setting. LEAP-002 is a phase 3 study which evaluates Lenvatinib with Pembrolizumab compared to Lenvatinib in patients with advanced HCC in the first-line setting [60]. The COSMIC 312 study is a randomized phase 3 trial evaluates Atezolizumab with Cabozantinib compared to Sorafenib as first-line systemic therapy for advanced HCC with a primary endpoint of PFS. Recently presented data from the COSMIC 312 study showed significant improvement in PFS in the Cabozantinib plus Atezolizumab arm compared to Sorafenib in treatment naive patients with advanced HCC, meeting its primary endpoint. Median PFS in the Cabozantinib plus Atezolizumab arm was 6.8 months compared to 4.2 months in the Sorafenib arm. OS was not met at time of interim analysis [61].

Chimeric antigen receptor T cell (CAR-T) therapy has shown promising results in hematologic malignancies and there are currently ongoing clinical trials to evaluate the role of CAR-T therapy in solid malignancies. CAR-T is a novel form of immunotherapy that allows T cells to recognize tumor-associated antigens (TAAs) [62]. Glypican-3 is a tumor-associated antigen specific to HCC and is currently being investigated as the target in multiple CAR-T clinical trials for HCC treatment [63].

## 7. Conclusions

HCC remains challenging to treat and hence a leading cause of cancer-related deaths. Barriers to treatment include advanced presentation of patients and concurrent advanced underlying liver disease. Major risk factors for HCC include conditions that lead to liver injury and cirrhosis including hepatitis infection, alcohol use, and non-alcoholic steatohepatitis. Screening for HCC is advised in patients with cirrhosis with imaging (typically ultrasound) as well as serum AFP. Determining treatment options for HCC is dependent on stage of disease, liver function, performance status, and size of tumors. Early-stage HCC is potentially curable with surgical resection and liver transplant. For localized advanced HCC options include liver-directed therapy such as transarterial chemoembolization [9].

The therapeutic landscape of systemic therapies for advanced, metastatic HCC has changed dramatically in recent years. Sorafenib was the only agent available for several years since its approval in 2007 but in 2022 there are multiple options for systemic therapy.

Atezolizumab/Bevacizumab is preferred as first-line systemic therapy in patients without contraindications to immunotherapy or VEGF inhibitors. For patients who are not candidates for Atezolizumab/Bevacizumab, therapy with tyrosine kinase inhibitors Sorafenib or Lenvatinib can be considered. Based on the promising results of the HIMALAYA, it is expected the combination of Durvalumab and Tremelimumab will be likely approved in the first-line setting. In the second-line setting, systemic therapy options include Cabozantinib, Regorafenib, Ramucirumab (in patients with AFP ≥ 400), Nivolumab, Pembrolizumab and Nivolumab/Ipilimumab. There are multiple ongoing clinical trials in early, intermediate and advanced stage HCC that will change the way HCC is treated in the future.

**Author Contributions:** Conceptualization, U.K.; formal analysis, A.S., S.Z., I.N., T.P. and S.U.; investigation, A.S., S.Z., I.N., T.P. and S.U.; resources, A.S., S.Z., I.N., T.P. and S.U.; data curation, A.S., S.Z., I.N., T.P. and S.U.; writing—original draft preparation, A.S., S.Z., I.N., T.P. and S.U.; writing— review and editing,, A.S., M.B. and U.K. supervision, U.K. and A.S.; project administration, U.K. and A.S.; All authors have read and agreed to the published version of the manuscript.

**Funding:** This research received no external funding.

**Conflicts of Interest:** The authors declare no conflict of interest.

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
