# Peer review of "A Review of Current and Emerging Therapies for Advanced Hepatocellular Carcinoma"

_curroncol, doi:10.3390/curroncol29090507_

Round 1

Reviewer 1 Report

Title: A review of current and emerging therapies for advanced hepatocellular carcinoma 

This review article summarize the key clinical trials that lead to approval of therapeutic agents for systemic treatment of hepatocellular carcinoma. This review article adds additional interpretation that would advance our understanding of the current and emerging therapies for advanced hepatocellular carcinoma. Overall, the manuscript is well-written and well-organized.

Author Response

Thanks, author revised.

Reviewer 2 Report

This study reviewed the key clinical trials that lead to approval of agents for systemic treatment of hepatocellular carcinoma (HCC), discuss the preferred sequence of treatment options, as well as prospective studies for management of HCC.

The explanation of each drug is detailed and easy to understand. In addition, the sequence of treatment options is reasonable and agreeable. However, I think some modifications are needed as follows.

1. Abbreviations/Abbreviations/Acronyms are defined when they first appear in three sections: abstract, text, and first figure/table. When defined for the first time, acronyms/abbreviations/acronyms are added in parentheses after the writing, but are not thoroughly in the text.

For example,

Line 125 PD-1

Line 280 programmed cell death 1 receptor (PD-1) etc.

Please comfirm.

2. Line 359

Based on this trial, Pembrolizumab 200mg IV or 400mg IV every three or six weeks, respectively, was approved in patients previously treated with sorafenib.

Please instead “IV” to “intravenously”

3. Line 401-404

The phase III RESORCE trial enrolled patients with CP-A and Barcelona Clinic Liver Cancer Stage Category B or C HCC, with HCC progression on sorafenib, where participants were randomized to receive 160mg of regorafenib once daily for the first 21 days of each 28 cycle until disease progression or overwhelming toxicity.

In the RESORCE study, participation was limited to patients who tolerated to sorafenib (400 mg/day or more for at least 20 of the last 28 days). I think this point is important and needs to be stated.

4. Line 409

Please delete “(7)” and insert reference if need.

5. Table 1. Table2.

“Placebo" should be unified with "placebo”. Do not capitalize.

6. In Table 1, 6nd row, “Ramacirumab” might be a typing error, please check and revise it.

7. In Figure1, “Ramcirumab” might be a typing error, please check and revise it.

8. Line 534

EMERALD-1, LEAP-012, Checkmate 74W are all phase 3, randomized, placebo controlled, trials with primary endpoints of EMERALD-1 and LEAP-012 being PFS and the primary end point of Checkmate 74w being time to TACE progression.

What does “TACE progression” mean? Please indicate in more detail.

Author Response

Thanks, author revised.